# *Streptococcus gordonii*: Pathogenesis and Host Response to Its Cell Wall Components

**DOI:** 10.3390/microorganisms8121852

**Published:** 2020-11-24

**Authors:** Ok-Jin Park, Yeongkag Kwon, Chaeyeon Park, Yoon Ju So, Tae Hwan Park, Sungho Jeong, Jintaek Im, Cheol-Heui Yun, Seung Hyun Han

**Affiliations:** 1Department of Oral Microbiology and Immunology, School of Dentistry, Dental Research Institute, Seoul National University, Seoul 08826, Korea; okjpark7@snu.ac.kr (O.-J.P.); kwonykag@snu.ac.kr (Y.K.); chaeyeon@snu.ac.kr (C.P.); sdsfs819@snu.ac.kr (Y.J.S.); pth0102@snu.ac.kr (T.H.P.); sunghojeong13@gmail.com (S.J.); jintaek1@snu.ac.kr (J.I.); 2Department of Agricultural Biotechnology and Research Institute of Agriculture and Life Sciences, Seoul National University, Seoul 08826, Korea; cyun@snu.ac.kr; 3Institute of Green Bio Science Technology, Seoul National University, Pyeongchang 25354, Korea

**Keywords:** apical periodontitis, biofilm, cell wall components, infective endocarditis, inflammation, opportunistic pathogen, *Streptococcus gordonii*

## Abstract

*Streptococcus gordonii*, a Gram-positive bacterium, is a commensal bacterium that is commonly found in the skin, oral cavity, and intestine. It is also known as an opportunistic pathogen that can cause local or systemic diseases, such as apical periodontitis and infective endocarditis. *S. gordonii*, an early colonizer, easily attaches to host tissues, including tooth surfaces and heart valves, forming biofilms. *S. gordonii* penetrates into root canals and blood streams, subsequently interacting with various host immune and non-immune cells. The cell wall components of *S. gordonii,* which include lipoteichoic acids, lipoproteins, serine-rich repeat adhesins, peptidoglycans, and cell wall proteins, are recognizable by individual host receptors. They are involved in virulence and immunoregulatory processes causing host inflammatory responses. Therefore, *S.*
*gordonii* cell wall components act as virulence factors that often progressively develop diseases through overwhelming host responses. This review provides an overview of *S. gordonii,* and how its cell wall components could contribute to the pathogenesis and development of therapeutic strategies.

## 1. Overview of Streptococcus gordonii

In 1884, Rosenbach first termed the *Streptococcus* group from examining a man with suppurative lesions [1]. This genus is classified as Gram-positive, cocci or spherical, and clustered pairs or chains (Figure 1) [2]. They are homofermentative and facultative anaerobes, exhibiting negative catalase activity and forming no spores [1]. *Streptococci* are divided into three groups based on hemolysis patterns on blood agar plates: β-hemolysis (complete hemolysis), α-hemolysis (incomplete hemolysis), and γ-hemolysis (no hemolysis) [3]. Recently, phylogenetic approaches ultimately subdivided *Streptococcus* into eight groups consisting of mitis, sanguinis, anginosus, salivarius, pyogenic, mutans, downei, and bovis [3].

*Streptococcus gordonii* is commensal, non-pathogenic bacterium that is present in the human body, including the skin, oral cavity, upper respiratory tract, and intestine. It mainly resides on mucosal surfaces, such as the oral cavity, but also live in water, soil, plants, and food [4,5]. *S. gordonii*, a part of the α-hemolytic (viridans) sanguinis group, primarily inhabits the oral cavity of humans and animals [6]. However, it is also an opportunistic pathogen and can cause a variety of infectious diseases (Figure 2). Recently, metagenomic next-generation sequencing analysis showed that *S. gordonii* exists in patients with apical periodontitis or caries and the heart valves of patients with infective endocarditis [7,8,9]. In fact, *S. gordonii*, an initial colonizer on the tooth surface, can co-aggregate with several other oral microorganisms, contributing to the development of periodontal disease and caries [10]. *S. gordonii* can enter the blood stream by oral bleeding, leading to endocarditis [6]. Furthermore*, S. gordonii* binds to the cell surface of various host cells, contributing to the initiation of diseases through the inflammatory responses [11,12].

The bacterial cell wall plays a crucial role in the survival and growth of bacteria [13]. In Gram-positive bacteria, the cell wall is composed of thick peptidoglycan (PGN) and various components including lipoteichoic acid (LTA), wall teichoic acid (WTA), cell wall-anchoring glycoproteins, and lipoproteins [14]. Cell wall components can be recognized by host receptors such as pattern recognition receptors (PRRs), which initiate host innate immune responses [14]. *S. gordonii* expresses cell wall proteins, including Streptococcal surface protein (Ssp) A, SspB, collagen-binding domain protein (CbdA), and serine-rich repeat (SRR) glycoproteins, such as gordonii surface protein B (GspB) and Hs antigen (Hsa) [15,16]. The cell wall proteins of *S. gordonii* easily adhere to platelets, erythrocytes, monocytes, and dendritic cells (DCs) that could lead to acute immune responses in humans [16,17]. Therefore, understanding how cell wall components of *S. gordonii* interact with host cells is required not only to determine its entire pathogenesis but also to apply for the treatment and prevention of *S. gordonii*-mediated diseases.

## 2. Diseases Associated with *S. gordonii*

### 2.1. Apical Periodontitis

Apical periodontitis is an inflammatory disease that occurs within periapical tissues [18]. A study in the U.S. has revealed that 4.1% of randomly sampled teeth have apical periodontal disease, and this rate increases up to 31.3% from teeth receiving endodontic treatment [19]. Interestingly, S. *gordonii* was predominantly isolated from 34 of 100 patients with apical periodontitis [10]. Apical periodontitis is mainly caused by bacterial invasion into the dental pulp and endodontic lesions, and the attachment of bacteria on dentin surfaces is a critical step for the development of apical periodontitis [18]. *S. gordonii* expresses numerous cell wall proteins that facilitate attachment on dentin surfaces [20]. After successful attachment, *S. gordonii* forms biofilm matrix by deposition of extracellular polysaccharide, which is an essential structural component of biofilm [21]. *S. gordonii* found in biofilms is more resistant to antibacterial agents than in the planktonic state, making it harder to eliminate *S. gordonii* [22,23]. Furthermore, the biofilm formation of *S. gordonii* on dentin surfaces facilitates invasion into the dental pulp through dentinal tubules [24]. In addition, *S. gordonii* can interact with host cells followed by its penetration into root canals or dentinal tubules [18].

The interaction subsequently induces inflammatory conditions on periapical lesions. For instance, *S. gordonii* induces the expression and secretion of chemotactic cytokines called interleukin (IL)-8 by stimulating toll-like receptor (TLR) 2 of human periodontal ligament cells (PDLs) [25]. In fact, *S. gordonii* lipoprotein triggers the secretion of IL-8, monocyte chemoattractant protein (MCP)-1, cyclooxygenase-2, and prostaglandin E2 in the human dental pulp cells (Figure 3) [26]. Moreover, several studies also demonstrated that *S. gordonii* induces the expression of IL-8 in human endothelial cells [27]. Consequently, *S. gordonii* can promote the infiltration of neutrophils and monocytes, causing acute inflammation on penetrated periapical lesions coincident with the activation of other host cells [28,29]. The presence of pro-inflammatory cytokines has been reported in apical periodontitis lesions [30,31]. Moreover, *S. gordonii* contributes to alveolar bone destruction, a common feature of apical periodontitis, through the upregulation of osteoclast differentiation or the downregulation of osteoblast differentiation [32]. *S. gordonii* induces pro-inflammatory cytokines such as IL-1β, IL-6, and IL-8, and these cytokines potently activate the differentiation of macrophages into osteoclasts [15,32]. In addition, it has been reported that *S. gordonii* directly activates the bone resorbing ability of osteoclasts and inhibits the bone forming activity of osteoblasts [29]. Therefore, *S. gordonii* is considered as an etiological agent that contributes to the development of apical periodontitis and alveolar bone resorption through the induction of acute inflammatory conditions.

### 2.2. Infective Endocarditis

*S. gordonii*, released from oral biofilms by tooth brushing, tooth extraction, or oral trauma, can disperse into the circulatory system through blood vessels, leading to systemic infections [22,33]. Infective endocarditis is a life-threatening disease caused by oral *streptococci* where the hospital mortality rate from this disease counts approximately 20% [34,35]. A nationwide analysis reported that infective endocarditis incidence in the U.S. recorded 47,134 cases [36]. Moreover, *S. gordonii* showed the highest prevalence rate for Streptococcal infective endocarditis in Denmark, which was 44.2% [37]. The persistent exposure of *S. gordonii* by low-grade bacteremia can cause infective endocarditis [38]. Therefore, an understanding of the pathogenesis of *S. gordonii* is necessary for the treatment and prevention of infective endocarditis. Interaction between *S. gordonii* and host cells in the bloodstream is considered as an important initial step in the pathogenesis of infective endocarditis [39]. When *S. gordonii* enters the bloodstream, *S. gordonii* preferentially binds to platelets or erythrocytes using their numerous cell surface proteins and then hematogenously spreads to damaged heart valves [40]. *S. gordonii* potently binds to human vascular endothelial cells and accumulates on heart valves to form biofilms [22,41], causing the aggregation of platelets that develops into bacterium-platelet-fibrin complexes and further exacerbates the inflammatory responses. In addition, *S. gordonii* activates human valve interstitial cells to induce IL-6 and IL-8, leading to the infiltration of immune cells through nuclear factor-kappa B (NF-κB) signaling pathway [42]. Immune cells in heart lesions recruited by chemokines are directly activated by *S. gordonii*. For instance, *S. gordonii* induces nitric oxide production on macrophages through TLR2 signaling pathway (Figure 3) [43]. Human monocytes stimulated with *S. gordonii* enhance pro-inflammatory cytokine production and express more cell surface markers, including cluster of differentiation (CD)40, CD54, and CD80 [44]. Moreover, *S. gordonii* stimulates DCs to cause the induction of pro-inflammatory cytokines, such as IL-6, IL-12, and tumor necrosis factor-α (TNF-α) and co-stimulatory receptors [16,45]. Taken together, *S. gordonii* originating from oral biofilms enters into the blood stream and induces platelet aggregation and excessive inflammatory conditions by stimulating various host cells.

### 2.3. Other Diseases

Once *S. gordonii* enters the bloodstream, it can translocate into other organs through the blood circulation system [46]. Therefore, *S. gordonii* can cause not only infective endocarditis but also other systemic diseases. For instance, when *S. gordonii* translocates into the joint, it can rapidly induce excessive inflammatory condition. The inflammatory responses induced by *S. gordonii* infections directly cause septic arthritis, leading to joint destruction and bone loss [33]. In addition, *S. gordonii* can cause empyema in the lungs, which is defined as an accumulation of pus in the pleural cavity. *S. gordonii* has been found in the pleural fluid of empyema patients [47]. *S. gordonii* has also been isolated in the perihepatic collection fluid originating from perihepatic abscesses that form around the liver [48]. Moreover, *S. gordonii* translocated into spinal bones induces inflammation, further developing into pyogenic spondylitis or spondylodiscitis [49,50].

## 3. Virulence Factors

### 3.1. Serine-Rich Repeat Adhesins

SRR adhesins are Gram-positive bacterial cell wall-anchored glycoproteins [51]. Among the SRR groups, Hsa and GspB are well-characterized homologous adhesins in *S. gordonii* strains CH1 and M99, respectively [52,53]. SRR adhesins include an N-terminal signal peptide, a short SRR1 region, a ligand-binding basic region (BR), a long SRR2 region, and a LPXTG motif at a C-terminal for cell wall anchoring [54,55,56]. Both Hsa (203 kDa) and GspB (286 kDa) include the basic SRR adhesion domains: an N-terminus, two SRRs, a BR, and a C-terminus (Figure 4) [52,53]. When *S. gordonii* expresses SRR adhesins, the accessory Sec system, which includes SecA2 and SecY2, mediates the exporting of SRR adhesins from bacterial cytoplasm to the cell wall [57]. Despite the fact that SRR adhesins contain conserved domains, the ligand-binding BR exhibits various structural changes in amino acid sequences, contributing to the binding specificity to each respective ligand [17,55]. For instance, while both Hsa and GspB bind to sialyl-T antigen, only Hsa can bind to 3′-sialyllactose due to the amino acid differences in the BR domains [58,59].

The binding of *S. gordonii* to host cells or tissues using various adhesins is an important step for the initiation of infection [46,60,61,62]. Among the various adhesins, SRR adhesins are considered as the major molecules for the development of *S. gordonii* pathogenesis (Table 1). There have been several studies on the attachment of *S. gordonii* to various host cells using SRR adhesins. For instance, *S. gordonii* binds to sialylated carbohydrate moieties of glycoprotein Ibα on platelet membranes through the BR domains of Hsa and GspB [17,40,57], and *S. gordonii* lacking GspB reduces binding potentials to rat and human platelets by approximately 65% [46,58]. In addition, Deng et al. reported that *S. gordonii* preferentially interacts with sialoglycans in platelets through the BR domain of SRR adhesins in whole blood [40]. Moreover, SRR adhesins of *S. gordonii* also mediate binding to α2-3-linked sialic acid on glycophorin A of erythrocyte membranes. Two arginine residues (Arg340 and Arg365) of Hsa are major sites of interaction with glycoproteins on erythrocyte membranes [59,63]. In catheterized animal models infected with *S. gordonii*, bacterial vegetation density challenged with wild type strain was higher than infected with Hsa- or GspB-deficient strain [39,46]. These accumulating studies indicate that SRR adhesins are responsible for the translocation of *S. gordonii* to the endocardium by attaching to circulating platelets or erythrocytes in the bloodstream causing the pathogenesis of infective endocarditis [39,40].

The adherence of *S. gordonii* by SRR adhesins to immune cells, such as polymorphonuclear leukocytes (PMNs), macrophages, DCs, or monocytes, evokes the promotion and activation of the host inflammatory condition [16,66,67,68]. Hsa specifically interacts with sialoglycoproteins on monocyte membranes such as CD11b, CD43, and CD50. In addition, mucin-like domains of glycoproteins on PMN surfaces are major sites of Hsa adhesion [66,68]. The interaction of HL-60 cell-derived monocytes with *S. gordonii* rapidly induces the expression of DC markers including CD1a, CD83, CD86, and IL-12 while Hsa-deficient *S. gordonii* fails to differentiate monocytes into DCs [67]. Moreover, compared to the wild type treatment, SRR adhesin-deficient *S. gordonii* poorly induces pro-inflammatory cytokines, including TNF-α, IL-6, and IL-12, in human monocyte-derived DCs coincident with attenuating the proliferation and activation of co-cultured T cells. These results indicate that SRR adhesins have the potential of modulating innate and adaptive immune responses [16].

Accumulating studies demonstrate that SRR adhesins are closely involved in the biofilm formation of some Gram-positive bacteria [69,70,71]. SRR adhesins of *S. gordonii* also play important roles in biofilm formation. Kim et al. reported that GspB-deficient *S. gordonii* forms less biofilm than the wild type strain on human dentin surfaces, suggesting that GspB is crucial for the development of dental biofilm formation [64]. In addition, Hsa-deficient *S. gordonii* exhibits lower biofilm formation than the wild type strain on plates coated with saliva, fetuin, or mucin [65].

### 3.2. Cell Wall Proteins

*S. gordonii* has a variety of virulence factors on their cell wall, such as antigen-related proteins, collagen-binding proteins, fibronectin-binding proteins (Fbps), platelet adherence proteins, and α-amylase-binding proteins. As summarized in Table 2, they are well-known to promote the species-specific binding of *S. gordonii* to various receptors, contributing to the development of dental caries, periodontal diseases, and endodontic diseases caused by *S. gordonii* [72].

The cell wall proteins, SspA and SspB, are antigen I/II family polypeptides of *S. gordonii* that mediate the attachment to various hosts and other bacterial cells [72]. These proteins accelerate the infection of *S. gordonii* into root dentinal tubules by binding type I collagen or β1 integrin, mediating the aggregation and adherence of cells by binding to salivary agglutinin glycoprotein (gp340), and facilitating biofilm formation by interacting with other oral bacterial species, such as *Porphyromonas gingivalis* [24,72,73]. *S. gordonii* also has antigen-related polypeptides, CshA and CshB. These antigen-related proteins are responsible for binding to host fibronectins or other oral microorganisms and facilitating invasion into endothelial cells [74,75].

CbdA is a protein of *S. gordonii* similar to *Enterococcus faecalis* collagen-binding protein, Ace, in the structure and function [77]. An in silico analysis of *S. gordonii* Cbd locus showed that CbdA contains a signal sequence at the amino terminus and LPXTG, a PGN anchor motif at the carboxyl terminus [77]. This protein promotes the attachment of *S. gordonii* to host type I collagen, suggesting that *S. gordonii* persists in its survival in instrumented root canals by CbdA [77].

FbpA is one of the bacterial cell wall proteins of *S. gordonii*. Fibronectin is a eukaryotic glycoprotein that exists in the plasma or on host cell surface and engages in cellular adhesion, migration, and differentiation. Fibronectin can also be a target molecule for bacterial attachment [60]. For example, FbpA affects the binding of *S. gordonii* to eukaryotic fibronectin through the regulation of CshA expression [60].

Platelet adherence protein A (PadA) is a protein that mediates interactions between the major platelet receptor GPIIb/IIIa and *S. gordonii* [79]. PadA activates platelets and promotes biofilm formation by cooperating with Hsa [78]. PadA and Hsa are involved in each other’s active presentation on the cell wall, indicating that they cooperatively mediate the activation of platelets and promotion of biofilm formation [78].

*S. gordonii* produces α-amylase-binding protein A and B (AbpA and AbpB) that contribute to biofilm formation and colonization on teeth. AbpA and AbpB bind to α-amylases secreted from animals, which may facilitate the binding of bacteria to the salivary pellicle [80]. In addition, they provide *S. gordonii* with nutritional benefits by capturing host enzymatic activity to compete with other oral microbial species [80].

Collectively, the cell wall proteins of *S. gordonii* are important for the interaction with host proteins (Figure 5). Therefore, further studies are necessary to fully understand the sophisticated functions of *S. gordonii* cell wall proteins and to control various *S. gordonii*-mediated infectious diseases.

### 3.3. Lipoproteins

Bacterial lipoproteins are located on the extracellular surface of the cytoplasmic membrane of bacteria. In general, Gram-positive and Gram-negative bacteria contain diacylated- or triacylated- lipoproteins, respectively [81]. Lipoproteins play various physiological functions, such as nutrient acquisition, adherence, adaptation to environmental changes, protein maturation, bacterial growth, and pathogenesis [82,83,84]. Therefore, the deletion of lipoprotein diacylglycerol transferases (*lgt*) gene, associated with the maturation of lipoproteins [85], causes changes in bacterial physiological properties of *Streptococci*. For example, the *Streptococcus pneumoniae lgt* mutant exhibits reduced growth in cation-depleted medium and reduced intracellular concentrations of several cations, such as Fe^2+^, Zn^2+^, and Mn^2+^ [86]. Moreover, mutation in *lgt* results in impaired growth and attenuated virulence in *Streptococcus sanguinis* [87]. On the other hand, there are no bacterial morphological, size, and growth pattern differences between lipoprotein-deficient *S. gordonii* and its wild type [88] (Table 3).

TLRs are one of the PRRs that can recognize bacterial pathogen-associated molecular patterns and are located on the surface or intracellular compartments of host cells [91]. *S. gordonii* possesses diacylated-lipoproteins, which are considered as more potent stimulators of TLR2 than LTA in *S. gordonii* [43,88]. Through TLR2 activation, lipoproteins can induce the secretion of pro-inflammatory cytokines and chemokines (Figure 3 and Table 3). For example, the recognition of diacylated- or triacylated lipoproteins, which are agonists of TLR2/TLR6 or TLR2/TLR1, respectively, triggers the MyD88-dependent signaling pathway [92]. In addition, it has been reported that the lipoproteins of *S. gordonii* induce the production of nitric oxide by activating the NF-κB pathway, STAT1 phosphorylation, and interferon (IFN)-β expression in RAW 264.7 cells [43]. Moreover, the lipoprotein-deficient mutant of *S. gordonii* fails to induce pro-inflammatory cytokines, such as TNF-α, IL-8, and IL-1β at both mRNA and protein levels in the human monocytic cell line THP-1, and mouse bone-marrow derived macrophages [88]. Lipoproteins of *S. gordonii* also induce TNF-α, IL-6, IL-12p70, and IL-10, and can upregulate the expression of the DC surface marker CD80 but not CD86 on bone-marrow DCs [89]. Wild type strain, but not lipoprotein-deficient strains, reduces the frequency of CD4^+^, CD25^+^, and Foxp3^+^ regulatory T cells in murine acute phase infection models [90]. Furthermore, the *S. gordonii lgt* mutant strain is cleared more rapidly in blood and organs such as the spleen and liver of mice than the wild type. Wild type *S. gordonii* strongly adheres to human umbilical vein endothelial cells compared to the *S. gordonii lgt* mutant strain [41]. Human PDLs exposed to purified lipoproteins of *S. gordonii* induce IL-8 production through the TLR2-mediated mitogen-activated protein kinase pathway [25]. Likewise, human dental pulp cells express pro-inflammatory mediators like IL-8 and MCP-1 when stimulated by *S. gordonii* lipoproteins [26]. Some research have revealed that Pam2CSK4, which mimics Gram-positive bacterial lipoproteins, induces increased expression of pro-inflammatory cytokines and immunosuppressive cytokines in human odontoblast-like cells [93]. Therefore, lipoproteins are involved in the virulence and immunoregulatory process, in which targeting the biosynthesis of lipoproteins could be a potential vaccine strategy [94].

### 3.4. Teichoic Acids

Teichoic acid (TA) is a predominant cell wall component of Gram-positive bacteria that accounts for approximately 50% of cell wall dry weight [95], and is distinctly classified by structure and location. TA is attached to the cell membrane (LTA) or to the cell walls (WTA). WTA is a PGN-linked polymer with 40 to 60 polymer repeats [96]. LTA is classified by polymeric chain diversity (types I to V) [97]. Among them, the structure of type I LTA is well-characterized. Type I LTA contains an unbranched 1–3 linked glycerol phosphate backbone structure anchored by glycolipids [97]. *S. gordonii* has type I LTA containing glycerol phosphate and terminal glucose-glycerol phosphate repeat units (Figure 6) [98].

LTA plays a critical role in cell division, growth, and biofilm formation. Lima et al. reported that LTA-deficient *S. gordonii* strain grows more slowly than wild type *S. gordonii* on solid media and exhibits decreased biofilm formation on saliva-coated hydroxyapatite disks [98]. In addition, they have analyzed the protein profiles of cell wall fractions from the wild type and LTA-deficient *S. gordonii* strains using mass spectroscopy. Acetate kinase, phosphoglycerate kinase, serine protease, SspA, and SspB are more abundant in LTA-deficient *S. gordonii* than the wild type strain. On the other hand, the cell division transport system ATP-binding protein, FtsE, and penicillin-binding protein 2a were more abundant in the wild type strain. These results suggest that LTA affects cell division, cell wall biosynthesis, and the biofilm formation of *S. gordonii*.

Accumulating reports suggest that LTA induces inflammatory responses through TLR2. *Lactobacillus plantarum* LTA induces nitric oxide production in the presence of IFN-γ in macrophages [99]. In addition, LTA of *Enterococcus faecalis* or *Streptococcus mutans* significantly increases inflammatory cytokines such as TNF-α and nitric oxide through TLR2 activation [100,101]. However, the induction of IL-8 or nitric oxide in human PDLs and murine macrophages by LTA-deficient *S. gordonii* is similar to the level expressed by wild type *S. gordonii* [25,43]. This finding indicates that, unlike LTA purified from other bacteria, *S. gordonii* LTA alone may not induce host immune responses, in other words, has less immunostimulating potential. In addition, immunomodulatory effects of LTA during inflammation have been reported. *L. plantarum* LTA efficiently inhibits Poly I:C-induced IL-8 production in porcine intestinal epithelial cells and lipopolysaccharide (LPS)-induced endotoxin shock [102,103]. *Staphylococcus aureus* LTA attenuates LPS-induced B cell proliferation [104]. Therefore, based on these results, future studies are needed to identify the immunomodulatory effects of *S. gordonii* LTA.

Like LTA, WTA affects various bacterial functions such as adhesion, growth, autolytic activity, and antibiotic resistance [96,105]. WTA also has been shown to modulate the maturation and activation of DCs [106]. DCs treated with WTA-deficient *S. aureus* exhibit lower DC maturation and activation than wild type *S. aureus*. However, to the best of our knowledge, no study has reported the effects of *S. gordonii* WTA in bacterial function and immunemodulation.

### 3.5. Peptidoglycan

PGN is one of the most abundant microbial cell wall components. It is a polymer of cross-linking linear chains composed of alternating N-acetyl glucosamine and N-acetyl muramic acid (NAM). Each NAM has a short peptide consisting of alternating _L_- and _D_-amino acids that involve in the cross-linking of linear sugar chains [107]. These peptides provide significant diversity of PGNs, which vary from species to species [107]. Although species-specific structures of some bacterial PGNs, such as *Mycobacterium* have been discovered [108], little is known about *S. gordonii* PGN. Due to the diversity of glycan strands and peptide moieties [109,110], it is expected that PGN structures vary from bacteria to bacteria. Therefore, identifying host responses triggered by *S. gordonii*-specific PGN would further elucidate the properties of *S. gordonii*.

Nucleotide oligomerization domains (NODs), NOD1 and NOD2, are mammalian intracellular proteins that engage in sensing bacterial PGN fragments [111]. It has been demonstrated that they recognize distinct bacterial PGN motifs. NOD1 specifically binds to Tri-diaminopimelic acid motifs of Gram-negative bacteria [111]. In contrast, NOD2 detects muramyl dipeptide motifs, suggesting its binding affinity for both Gram-positive and Gram-negative bacterial PGNs [112]. The detection of bacterial PGNs through NOD1 and NOD2 triggers pro-inflammatory responses by activating the NF-κB pathway, which is essential for bacterial clearance from infected host cells [113]. For example, a previous report has demonstrated the synergistic effects of TLR2 and NOD2 signaling pathways on IL-8 production in human PDLs [114]. Another report has revealed that cell wall components, LTA, lipoprotein, and PGN from *S. gordonii* induces the secretion of pro-inflammatory cytokines such as IL-6 and TNF-α in DCs through the TLR2 signaling pathway [89]. However, specific mechanisms or roles of *S. gordonii* PGN are poorly studied, and thus further mechanistic studies are necessary.

## 4. Therapeutic Strategies against *S. gordonii* Infections

Nowadays, a large number of therapeutics are available to avoid *S. gordonii*-induced oral and systemic diseases. For example, several antibacterial agents, such as fluorides, chlorohexidine (CHX), and antibiotics, have been widely used to regulate bacterial growth and infections [115,116,117,118]. However, those conventional antibacterial agents can affect bacterial viability or growth, imposing higher selective pressure that adversely increases the possibility of resistance development [119]. In addition, once bacterial biofilm is formed, the therapies are difficult to function properly due to reduced bacterial sensitivity [120]. Therefore, novel therapeutic strategies that target major virulence factors, virulence-mediated pathways, and biofilm formation could be a promising alternative to conventional therapies such as antibiotics.

As mentioned previously, it is well-known that *S. gordonii* binds to human platelets through the surface proteins Hsa and PadA, which is considered as a crucial step for infective endocarditis [121,122]. Based on this, one study revealed that antibodies targeting Hsa and PadA could delay or even inhibit the platelet aggregation of *S. gordonii* [123]. Several authors suggest that using these antibodies is likely to be a novel *S. gordonii* therapeutic strategy [123]. However, no follow-up studies have been made towards the development of *S. gordonii*-specific vaccines or treatments. Therefore, more research should be performed in various ways, as suggested in the following.

### 4.1. Regulation of Biofilm

There are many conventional methods to regulate biofilm. Especially for oral biofilm, physical removalby tooth brushing and scaling, is one of the most effective strategies to disrupt the biofilm. Several studies have demonstrated that tooth brushing results in an attenuation of plaque levels [124,125]. In addition, many other studies have revealed that various techniques of scaling, such as hand and ultrasonic scaling, laser scaling, and chemical scaling, have been used to control plaques [126,127].

Another traditional method to control biofilm is through the use of various chemicals, such as CHX, sodium chloride, and calcium chloride. CHX is a potential antimicrobial agent, which participates in disrupting biofilms by binding to negatively charged bacterial cell walls [128,129,130]. Salts, such as sodium chloride and calcium chloride, may weaken electrostatic interactions and, therefore, can attenuate the biofilm matrix [131].

*S. gordonii* and other oral *Streptococci* are well-known initial colonizers on tooth surfaces since they generate abundant cell wall adhesins interacting with various types of cells and tissues [132,133,134]. Therefore, targeting such surface colonization by disrupting the adherence of *S. gordonii* could likely be an effective therapeutic strategy against *S. gordonii* infections. The target candidates for these therapies can be SRRs and cell wall proteins because they are the major binding factors of *S. gordonii*. For example, the attachment and internalization of *S. gordonii* to epithelial cells occurs with SspA and SspB (antigen I/II family) [135]. The SRR adhesins of *S. gordonii* such as GspB are important for biofilm formation and bacterial aggregation on the surfaces [64]. Additionally, CshA, a multifunctional fibrillary adhesin, binds to host fibronectin and mediates the colonization of *S. gordonii* [11]. Therefore, the development of agents that can block these proteins can be an effective way to inhibit bacterial adherence and subsequent infections.

Using small molecules that suppress quorum sensing-mediated bacterial communication is in the spotlight as a novel therapeutic strategy for regulating biofilm. By using quorum sensing inhibitors, it is expected biofilm maturation can be easily regulated and bacterial self-protection attenuated from antimicrobial agents furnished by a mature biofilm [136,137]. Multiple approaches could be taken for inhibiting quorum sensing. The most promising approach to regulate quorum sensing involves targeting the S-ribosylhomocysteine lyase (LuxS)/autoinducer-2 (AI-2) quorum sensing system, one of the major bacterial communication systems of *S. gordonii* [138,139]. Therefore, using quorum-sensing inhibitors associated with LuxS/AI-2 could be an effective alternative to conventional therapies.

Promoting biofilm dispersion is another attractive approach for specific therapies against biofilm-mediated diseases, which can contribute to overcoming bacterial resistance mechanisms [140]. Several studies have revealed that bacterial LTA has an anti-biofilm effect, such as inhibiting biofilm formation, disrupting preformed biofilm, and consequently increasing susceptibility to antibiotics [141,142,143]. However, conversely, planktonic bacteria after biofilm disruption will likely disperse into the entire body and may cause issues in other sites of the host. Therefore, using conventional antibiotics together with LTA would be an effective way for the complete removal of infected microbes [140].

During bacterial infection, host cells recognize the invaded bacteria through PRRs and produce inflammatory mediators, such as antimicrobial peptides, immunoglobulin (Ig), and nitric oxide [14]. Some studies have revealed that inflammatory mediators can directly inhibit biofilm formation. For instance, human β-defensin, which is an antimicrobial peptide, decreases not only single species bacteria biofilm but also multi-species bacteria biofilm through the promotion of bacterial cell death [144,145]. In addition, among the Ig family, only secretory IgA, which is abundant in the oral cavity, effectively inhibits bacterial biofilm formation [146]. On the other hand, nitric oxide also inhibits biofilm formation by triggering bacterial dispersion [147]. Therefore, the activation of host cells to produce inflammatory mediators could be one of the alternative strategies for removing bacterial biofilm.

### 4.2. Inhibition of Bacterial Cell Wall Components

#### 4.2.1. Lipoprotein

Since lipoprotein-deficient *S. gordonii* remarkably reduces the induction of inflammatory responses [90], targeting lipoproteins can be one of the therapeutics used against *S. gordonii* infections. Lipoprotein structures react sensitively to changes in the environment. For example, *S. aureus* SitC lipoproteins usually exist in the diacyl form under acidic conditions [148]. In cases of *S. pneumonia*, lipoprotein-based vaccines and antibodies are being developed [149]. Pneumococcal lipoproteins such as pneumococcal surface adhesion A (PsaA), histidine triad protein (Pht) D, PhtE, PhtA, and PhtB are identified as the antigenic regions within Pneumococcal proteins. PsaA, in particular, has been shown to be immunogenic [150]. It has been reported that anti-PsaA monoclonal antibodies efficiently protect against *S. pneumonia* infections [151]. Therefore, lipoproteins of *S. gordonii* also can be a potential vaccine candidate.

#### 4.2.2. Lipoteichoic Acid/Wall Teichoic Acid

In general, it is well known that LTA is critical for cell division and growth. The binding of LTA to TLR2 triggers the host innate immune response. Notably, the platelet-activating factor receptor (PAFR) is essential for LTA-induced mucin gene expression in epithelial cells [152]. The inhibition of PAFR attenuates the LTA-induced expression of pro-inflammatory cytokines and chemokines but does not affect LPS-induced pro-inflammatory cytokines and chemokines [153,154]. These reports suggest that PAFR can be one of the potential targets against Gram-positive bacterial infections. Anti-LTA monoclonal antibodies have been reported to improve mice survival rates in methicillin- or vancomycin-resistant *S. aureus*-induced sepsis models [155]. Human serum anti-WTA IgG induces the opsonophagocytosis of *S. aureus* by PMNs [156]. In addition, Congo Red is a selective inhibitor for the *S. aureus* LTA synthase of *S. aureus*, which is thought to be a potential drug target against Staphylococcal infections [157]. However, because Congo Red is highly toxic [158], it would be desirable to use its molecular action mechanisms for the development of potential therapeutics. A positively charged molecule such as antimicrobial peptides like β-defensins may mask the immunostimulating effect of LTA by binding to LTA’s negative charge. Likewise, it is expected that specific molecule targeting LTAs or WTAs of *S. gordonii* can be developed as a therapeutic agent.

#### 4.2.3. Peptidoglycan

Since PGN is predominantly found in the bacterial cell wall, β-lactam antibiotics, such as penicillin, are used to induce bacterial cell death through blocking the linkage of PGNs [159]. In addition, vancomycin, an actinobacteria-derived glycopeptide, also inhibits the synthesis of PGNs by preventing transpeptidase and transglycosylase activities [160]. However, these traditional antibiotics can cause the emergence of antibiotic-resistant bacteria. Therefore, the discovery of new cell wall inhibition methods without promoting bacterial resistance has been investigated in recent studies. For example, Ling et al. reported that teixobactin, a new antibiotic, inhibits Gram-positive bacterial cell wall synthesis by binding to precursors of PGN or WTA [161]. Autolysins can be used as antimicrobial agents for breaking PGN bonds [162]. Furthermore, the use of bacteriophages instead of antibiotics is a renewed alternative strategy for the clearance of bacteria on lesions. Bacteriophages have endolysins or lysins that specifically target bacteria and break down the PGN bonds [163]. It has been reported that infective endocarditis was treated with lysins applied to the lesions [164]. Interestingly, PGN fragments can be released through the process of PGN breakdown and then be recognized by host cells through their NOD1 or NOD2 receptors [112]. The recognition of PGNs by NOD1 or NOD2 receptors triggers immune responses [113]. Therefore, PGN-specific receptors can be used as molecular targets for the clearance of *S. gordonii* through the activation of host immune cells.

## 5. Conclusions

*S. gordonii*, a commensal and opportunistic Gram-positive bacterium, plays an important role in the development of apical periodontitis and infective endocarditis. The attachment of *S. gordonii* to host cells using its adhesion proteins is an essential initiation step for disease development. Once attached, the cell aggregation of *S. gordonii* leads to biofilm formation on the apical lesions and heart valves. They also continuously interact with host cells, and the molecular interaction led by its cell wall components induces the immune responses. Therefore, *S. gordonii* may contribute to the progression of diseases due to its attachment, aggregation, and immunostimulatory effects. The bacterial cell wall is involved in important functions of bacteria, such as maintaining bacterial cell shape integrity, and contributing to bacterial growth, reproduction, metabolism, and movement. Therefore, by specifically inhibiting bacterial cell wall components, we could potentially alleviate or even prevent the bacterial infections. The development of novel anti-bacterial molecules, which is involved in targeting specific virulence factors of biofilms, is becoming more important in the prevention and treatment of various bacterial infections including *S. gordonii*-mediated diseases.

## Figures and Tables

**Figure 1 microorganisms-08-01852-f001:**
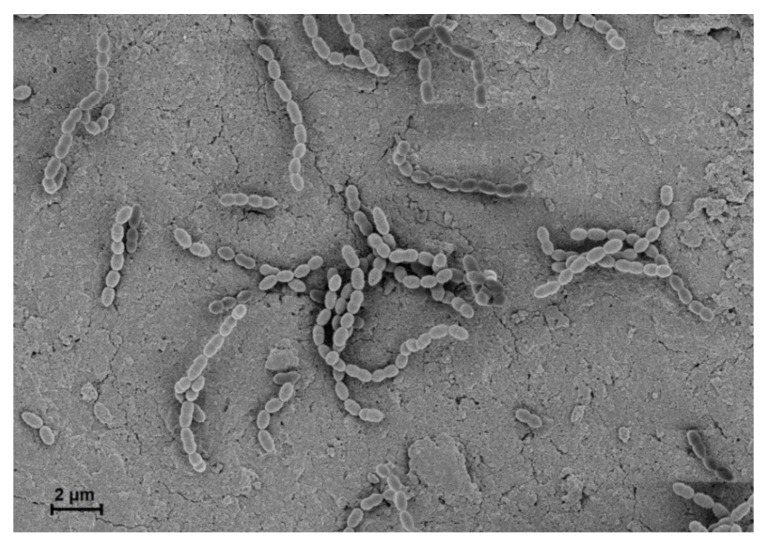
Scanning electron microscopic view of *S. gordonii* on a human dentin slice. *S. gordonii* was grown overnight in Todd Hewitt broth with 0.5% yeast extract at 37 °C and then diluted at 1:100 in the fresh medium. Human dentin slices were prepared from premolars with a single root. *S. gordonii* was grown on sterile dentin slices at 37 °C for 6 h. *S. gordonii* was visualized under a scanning electron microscope. *S. gordonii* exhibited spherical and clustered pairs or chains (10,000×). This experiment was performed under the approval of the Institutional Review Board of Seoul National University Dental Hospital, Seoul, Korea (CRI 17010).

**Figure 2 microorganisms-08-01852-f002:**
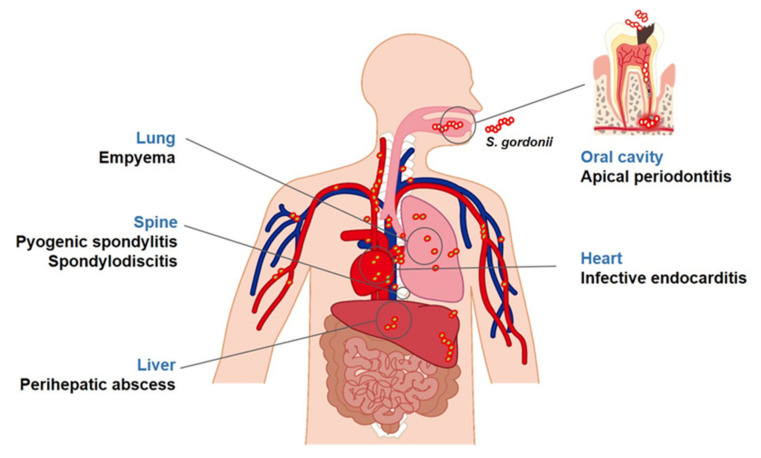
Diseases associated with *S. gordonii*. *S. gordonii*, a commensal bacterium, is found in areas of the human body, such as the oral cavity and skin. It is also an opportunistic pathogen associated with several diseases. In the oral cavity, *S. gordonii* is known to be closely associated with apical periodontitis. In addition, it can smear the blood stream during oral trauma and tooth extraction and disperse into various organs potentially causing systemic diseases including endocarditis, empyema, perihepatic abscesses, pyogenic spondylitis, or spondylodiscitis.

**Figure 3 microorganisms-08-01852-f003:**
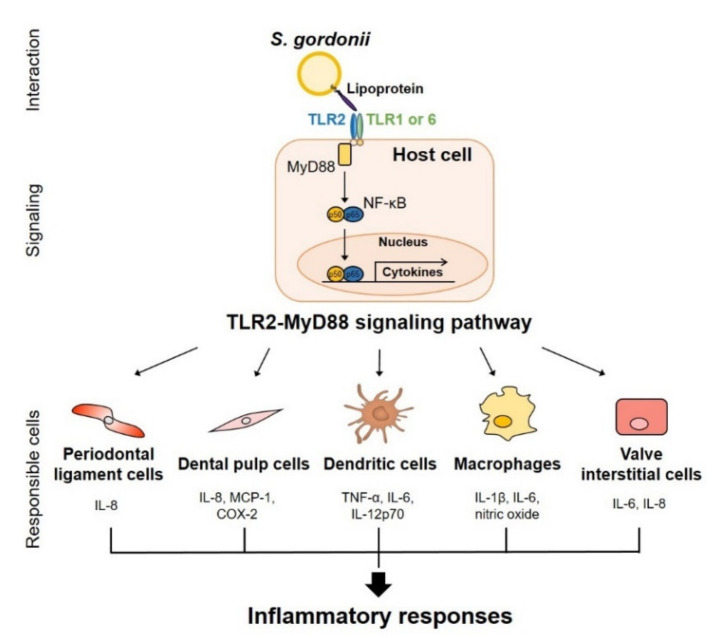
Inflammatory responses mediated through *S. gordonii* lipoproteins. Lipoproteins, which are the major virulence factor of *S. gordonii,* are directly recognized by heterodimers, consisting of toll-like receptor (TLR) 2 together with TLR1 or TLR6 on host cells, such as periodontal ligament cells, dental pulp cells, dendritic cells, macrophages, and valve interstitial cells. After activation of TLR2, myeloid differentiation primary response 88 (MyD88), an adaptor molecule of TLR2, mediates the activation of transcription factor nuclear factor-kappa B (NF-κB), leading to production of pro-inflammatory cytokines and chemokines, cell maturation, and infiltration of immune cells into lesions. These processes contribute to development of apical periodontitis or infective endocarditis by inducing inflammatory responses.

**Figure 4 microorganisms-08-01852-f004:**
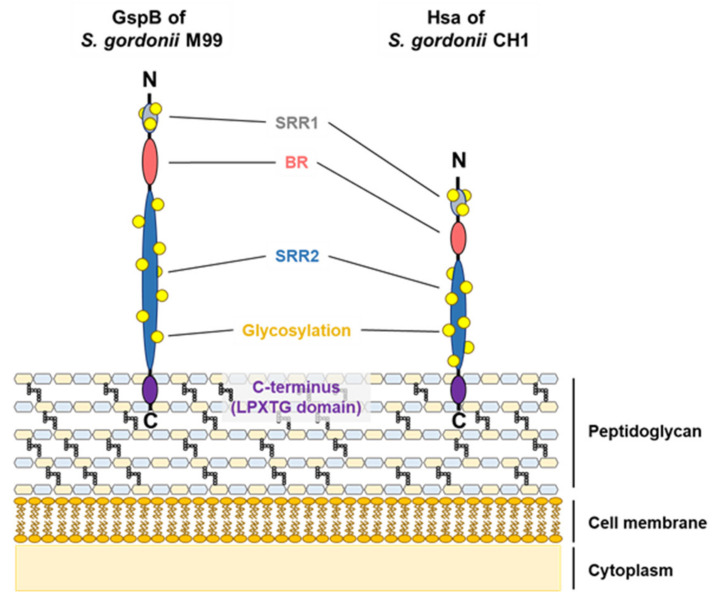
Illustration of the *S. gordonii* SRR structure. The serine-rich repeat (SRR) adhesins include a short SRR1 region (grey), a ligand-binding basic region (BR; red), a long SRR2 region (blue), and an LPXTG motif at a C-terminal (purple) for cell wall anchoring. Two SRR regions are highly glycosylated (yellow).

**Figure 5 microorganisms-08-01852-f005:**
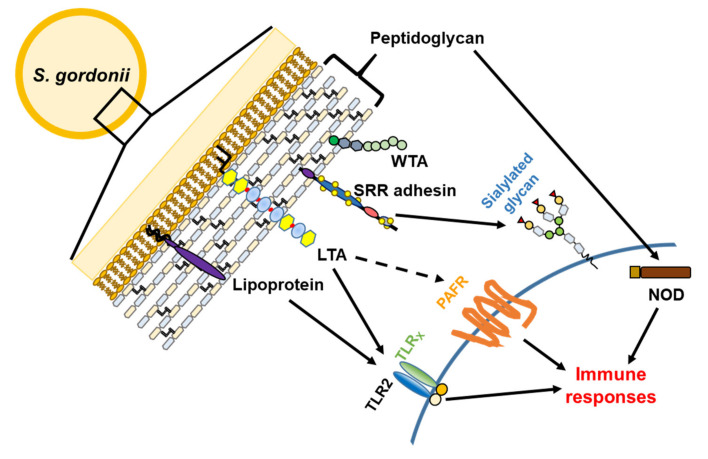
Major cell wall associated virulence factors of *S. gordonii* and their recognition receptors in the host. The cell wall of *S. gordonii* is mainly composed of peptidoglycans, lipoteichoic acid (LTA), wall teichoic acid (WTA), lipoproteins, and SRR adhesins. Lipoproteins and LTA are recognized by dimeric receptors containing TLR2 and TLRx (x can be 1, 2, 6, or 10). Platelet-activating factor receptor (PAFR) can be necessary for LTA recognition. SRR adhesins are important for the binding of *S. gordonii* to host cells through sialylated glycans. Peptidoglycans are recognized by an intracellular receptor, nucleotide oligomerization domain (NOD). The engagement of these cell-wall associated virulence factors stimulate the host innate immune responses through their specific receptors.

**Figure 6 microorganisms-08-01852-f006:**
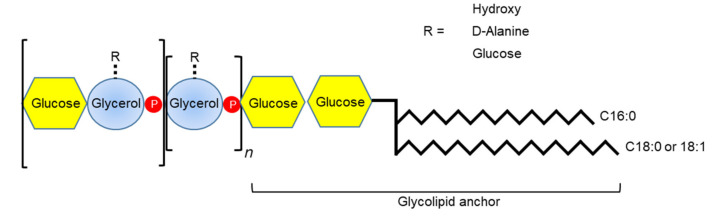
The structure of *S. gordonii* LTA. *S. gordonii* has type I LTA with a glycerol phosphate backbone structure anchored by glycolipid and terminal glucose-glycerol phosphate repeating units. The R site in the repeating units can be one of glucose, D-alanine, and hydroxyl groups though most of the R site is occupied with glucose. Fatty acids of *S. gordonii* LTA are heterogeneous containing C 16:0, C18:0, and C18:1.

**Table 1 microorganisms-08-01852-t001:** Serine-rich repeat adhesins of *S. gordonii* and their common features.

Target Surfaces	SRR Adhesins	Functions	References
Dentin slices	GspB	Facilitating binding of *S. gordonii* Contributing to biofilm formation and colonization on teeth	[64]
Plates coated with saliva	Hsa	Facilitating dental biofilm formation	[65]
Platelets	GspB Hsa	Mediating attachment to host cell membrane Providing opportunities of translocation to other organs from the oral cavity Contributing to the aggregating of platelets	[17,40,57]
Erythrocytes	Hsa		[59,63]
Polymorphonuclear leukocytes	Hsa	Promoting adherence to host cell surfaces	[66]
Monocytes	Hsa	Facilitating adherence to cluster of differentiation (CD)11b, CD43, and CD50 on the host cell membrane Inducing differentiation of monocytes into dendritic cells	[67,68]
Dendritic cells	GspB Hsa	Facilitating binding of *S. gordonii* Promoting induction of tumor necrosis factor-α (TNF-α), interleukin (IL)-6, and IL-12 production Activating maturation of dendritic cells	[16]

**Table 2 microorganisms-08-01852-t002:** Cell wall proteins of *S. gordonii* and their common features.

Cell Wall Protein		Host Binding Site	Functions	References
Antigen-related protein	SspA and SspB	Type I collagen or β1 integrin Salivary agglutinin glycoprotein (gp340)	Accelerating infection of *S. gordonii* into root dentinal tubules Mediating aggregation and adherence of cells Facilitating biofilm formation by interacting with other bacteria	[24,72,73]
	CshA and CshB	Fibronectin	Facilitating invasion into endothelial cells	[74,75]
Collagen-binding protein	CbdA	Type I collagen	Promoting *S. gordonii* to persist its survival in instrumented root canals	[76,77]
Fibronectin-binding protein	FbpA	Fibronectin	Controlling the attachment of *S. gordonii* to fibronectin through the regulation of CshA expression	[60]
Platelet adherence protein	PadA	Platelet receptor GPIIb/IIIa	Activating platelets by cooperating with Hsa Promoting biofilm formation by cooperating with Hsa to bind to salivary glycoproteins affecting Hsa active presentation on the cell wall	[78,79]
α-amylase-binding protein	AbpA and AbpB	α-amylase	Contributing to biofilm formation and colonization on teeth Facilitating the binding of *S. gordonii* to the salivary pellicle Providing nutritional benefit by capturing host enzymatic activity to compete with other oral microbial species	[80]

**Table 3 microorganisms-08-01852-t003:** Host immune responses by *S. gordonii* lipoproteins.

Host Cell	Stimuli	Receptor or Mechanisms	Reactions	References
**Murine macrophage**	*S. gordonii* lipoprotein extract	Toll-like receptor (TLR) 2-mediated nuclear factor-kappaB (NF-κB) pathway	Increased production of nitric oxide Reduced production of interferon-β (IFN-β)	[43]
**Human and mouse macrophage**	*S. gordonii* Δ*lgt*	TLR2-mediated NF-κB pathway	Failed to induce tumor necrosis factor-α (TNF-α), interleukin (IL)-8, IL-1β Reduced mortality	[88]
	*S. gordonii* lipoprotein extract		Increased expression of TNF-α, IL-8, and IL-1β	
**Murine dendritic cell**	*S. gordonii* lipoprotein extract	TLR2-mediated MyD88 pathway	Increased TNF, IL-6, IL-12p70, and IL-10 production Increased cluster of differentiation 80 expression	[89]
**Natural regulatory T cells in mouse splenocytes**	*S. gordonii* wild type and Δ*lgt*	TLR2-mediated NF-κB pathway	Only wild type reduced the frequency of natural regulatory T cells	[90]
**Human embryonic kidney 293 cell**	Heat-killed *S. gordonii* Δ*lgt*		Failed to induce NF-κB activation	
**Human vascular endothelial cells**	*S. gordonii* Δ*lgt*	TLR2-mediated NF-κB pathway	Weak adherence to the human umbilical vein endothelial cells	[41]
**Mouse tissues**			Rapid clearance from the blood flow, liver, and spleen Reduced amounts of TNF-α production	
**Human periodontal ligament cell**	*S. gordonii* lipoprotein extract	TLR2-mediated mitogen-activated protein kinase pathway	Increased production of IL-8	[25]
**Human dental pulp cell**	*S. gordonii* lipoprotein extract	TLR2-mediated NF-κB pathway	Increased mRNA expression level of IL-8 and monocyte chemoattractant protein-1	[26]

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
