# Peer review of "Streptococcus gordonii: Pathogenesis and Host Response to Its Cell Wall Components"

_microorganisms, 2020, doi:10.3390/microorganisms8121852_

Round 1
Reviewer 1 Report
The authors described pathogenesis and host response for S. gordonii especially in view of cell wall components including SRR adhesions, cell surface proteins, lipoproteins, teichoic acids and peptidoglycan. The authors gave an overview how S. gordonii infects tooth surfaces and heart valves and later explain the virulence factors on the cell surface, which are involved in infection and immune responses. Finally they described the therapeutic strategies against S. gordonii infection which would be helpful for drug developments to cure and prevent the bacterial infection. The review paper is well organized and well written. The paper is worth publication in Microorganisms.
Here are minor revisions.
- All the figures must be enhanced for their resolution. Especially Figure 1 and 5 are not clear when magnified.
- There is no citations of figures and tables in the main text.
3. In line 173: Deng et al., Line 199: Kim et al. et al. must be in italic.
4. In line 300: Glycerolphosphate must be glycerol phosphate or glycerophosphate.
5. Words that are not repeated are abbreviated. Check those in the whole text. For example, in line 313, acetate kinase appears only once, so (AckA) must be deleted.
6. Write down the full name (abbr.) where the abbreviations are first used.
7. Write down the experimental method in more detail in Figure 1 legend. You need to remove bottom letters which are not visible. For better visualization, you can give color contrast for the bacteira and explain it in the figure legend. - You used the human dentin slice, please give record whether IRB is approved or not.
Reviewer 2 Report
File is attached

Reviewer 3 Report
The topic of bacteria- host interaction is of great interest in the field. Park et al. have written a comprehensive review on Streptocuccus gordonii, in particular its pathogenesis and the host responses. I recommend the manuscript for publication in the journal with a minor revision. Please find issues/comments that can further strength the review as below.
One of the confusions throughout the manuscript is how authors distinguish planktonic and biofilm. According to the context, biofilm form is pathogenic, while the planktonic is commensal. But the virulence factors the manuscript discussed are more likely from the planktonic bacteria. Please clarify in the text and make it clear which form is authors refer to. Alternatively, separate into two sections for planktonic and biofilm.
The focus of the manuscript is the cell wall component, I am wondering if extracellular substances play a role in its pathogenesis. It would be better to include the section of EPS.
Please clarify cell wall, cell membrane and cell surface authors referred.
Figure1, intestine is mentioned in the caption, but not pointed out in the image. Please either highlight the intestine with the associated disease or remove intestine in the caption.
Table2, working with Has, be specific with “working”
It would be better to include a figure summarizing 2.1 and 2.2, to highlight the signaling pathways that S. gordinii stimulates. It will be a good reference to Table 3, receptor or mechanism.
Line 28, add comma before and
Line 39, Streptocuccus, italic
Line 199, start a new paragraph after endocarditis.
Line 120, considered as…
Line 221, how similar? Protein structure? Function?
Line 257-258, irrelevant information on S. aureus. Suggest removing it.
Line 426, 437, 453, please consider listing subject and remove control of.
